

# A comparison of fourteen fully characterized mammalian-associated *Campylobacter fetus* isolates suggests that loss of defense mechanisms contribute to high genomic plasticity and subspecies evolution

Susan A. Nadin-Davis[1], John Chmara[1], Catherine D. Carrillo[2], Kingsley Amoako[3], Noriko Goji[3], Marc-Olivier Duceppe[1] and John Devenish[1]

[1] Ottawa Laboratory Fallowfield, Canadian Food Inspection Agency, Ottawa, ON, Canada
[2] Ottawa Laboratory Carling, Canadian Food Inspection Agency, Ottawa, ON, Canada
[3] National Centre for Animal Diseases, Canadian Food Inspection Agency, Lethbridge, AB, Canada

Corresponding author
Susan A. Nadin-Davis,
nadindavis@gmail.com

## ABSTRACT

*Campylobacter fetus* is currently classified into three main subspecies, but only two of these, *C. fetus* subspecies *fetus* and *C. fetus* subsp. *venerealis* originate principally from ruminants where they inhabit different niches and cause distinct pathogenicity. Their importance as pathogens in international trade and reporting is also different yet the criteria defining these properties have never been fully substantiated nor understood. The situation is further compromised because the ability to differentiate between these two closely related *C. fetus* subspecies has traditionally been performed by phenotypic characterisation of isolates, methods which are limited in scope, time-consuming, tedious, and often yield inconsistent results, thereby leading to isolate misidentification. The development of robust genetic markers that could enable rapid discrimination between *C. fetus* subsp. *fetus* and subsp. *venerealis* has also been challenging due to limited differences in the gene complement of their genomes, high levels of sequence repetition, the small number of closed genome sequences available and the lack of standardisation of the discriminatory biochemical tests employed for comparative purposes. To yield a better understanding of the genomic differences that define these *C. fetus* strains, seven isolates were exhaustively characterised phenotypically and genetically and compared with seven previously well characterised isolates. Analysis of these 14 *C. fetus* samples clearly illustrated that adaption by *C. fetus* subsp. *venerealis* to the bovine reproductive tract correlated with increasing genome length and plasticity due to the acquisition and propagation of several mobile elements including prophages, transposons and plasmids harbouring virulence factors. Significant differences in the repertoire of the CRISPR (clustered regularly interspersed short palindromic repeats)-*cas* system of all *C. fetus* strains was also found. We therefore suggest that a deficiency in this adaptive immune system may have permitted the emergence of extensive genome plasticity and led to changes in host tropism through gene disruption and/or changes in gene

expression. Notable differences in the sub-species complement of DNA adenine methylase genes may also have an impact. These data will facilitate future studies to better understand the precise genetic differences that underlie the phenotypic and virulence differences between these animal pathogens and may identify additional markers useful for diagnosis and sub-typing.

## INTRODUCTION

The *Campylobacter* genus comprises a diverse and emerging group of Gram negative, curved and spiral rod shaped bacteria (*Fitzgerald, 2015*) currently divided into 44 distinct species by the List of Prokaryotic names with Standing in Nomenclature (LPSN-accessed April 17, 2020) (*Parte, 2018*) while potential new members continue to be described (*Silva et al., 2020*). In this genus, three subspecies of C. *fetus* are recognised, one of which, the genetically divergent C. *fetus* subsp. *testudinum* (*Gilbert et al., 2016*), is normally associated with reptilian hosts only (*Fitzgerald et al., 2014*) and will not be discussed further here (*Sprenger, Zechner & Gorkiewicz, 2012*). The other 2 subspecies, C. *fetus* subsp. *fetus* (CFF) and C. *fetus* subsp. *venerealis* (CFV), can inhabit both the gastrointestinal or reproductive tracts of ruminants including cattle (*Sprenger, Zechner & Gorkiewicz, 2012*). Strains of CFF have been reported as the cause of serious septicaemic infections in humans. This is observed especially in elderly patients or those with an underlying medical condition involving some form of immunodeficiency (*Gazaigne et al., 2008*; *Pacanowski et al., 2008*). CFV has been isolated from humans on rare occasions but its role in disease is only suspected because identification to the subspecies level is not conducted in many human infections caused by C. *fetus* (*Wagenaar et al., 2014*). The usual mode of transmission to humans is through ingestion or by injection of contaminated medical products. As such, CFF and CFV are considered to be accidental and opportunistic pathogens of humans that act as dead end hosts. It is believed that the fastidious nature of this microorganism results in significant underreporting of its prevalence and its role in disease (*Van Bergen et al., 2008*) and the serious nature of CFF, and possibly CFV, infections in humans makes them a public health concern (*Butzler, 2004*; *Woo et al., 2002*).

Regarding pathogenesis in ruminants, disease and host affiliation, CFF is found mainly in the intestinal tract of cattle, sheep and other ruminants and is believed to spread through the fecal/oral route by grazing on infected fields where the resulting ingestion back into the intestinal tract causes a descending systemic infection which localizes in the placental tissues in pregnant ewes and cows resulting in storm abortions in sheep and occasional abortion in cattle (*Sprenger, Zechner & Gorkiewicz, 2012*). CFV is found only in the reproductive tract of cattle but its classification is further complicated by the description of a distinct biovar, C. *fetus* subsp. *venerealis* biovar intermedius (CFVi) which, unlike CFV, is not confined solely to the genital tract of cattle but has also been isolated

from the intestinal tract (*Florent, 1963*). CFVi is aligned with CFV solely because of the same subspecies designation. In contrast to CFF, both CFV, and by association CFVi, are considered venereally transmitted through natural mating or by contaminated semen or equipment and, unlike CFF, produce an ascending, non systemic, infection in cows with the main manifestation of disease being infertility and, occasionally, early embryonic death leading to abortion in pregnancy. It is the condition of infertility and venereal spread caused by CFV and CFVi, not oral systemic spread and abortion caused by CFF, which defines the disease bovine genital campylobacteriosis (BGC). BGC is highly prevalent in some countries, including several in South America (*Silveira et al., 2018*), and is associated with high economic losses (*Sprenger, Zechner & Gorkiewicz, 2012*; *Veron & Chatelain, 1973*). In other countries, including Canada, reporting of this disease is rare but CFV and CFVi are occasionally isolated by culture screening of bulls serviced at artificial insemination (AI) facilities. As a result of the serious nature of BGC and its world-wide distribution, the World Organization for Animal Health (OIE) has designated BGC, and by association CFV/CFVi, a listed notifiable disease important in international trade (*OIE, 2018*) and this has not changed for many years.

However, gaps and contradictions in the scientific literature and knowledge regarding infertility and BGC exist and have remained for as long as BGC has been described. The role of CFVi in ruminant disease has never been truly elucidated even in official reference texts (*OIE, 2018*). In addition, attempts to verify infertility using manual introduction of CFV into the vulva of cows failed with no attempt to test CFF strains in parallel (*Clark, 1971*). Other studies showed that both type A ($H_2S$ and 1% Glycine growth tolerance negative resembling CFV) and type B strains ($H_2S$ positive and 1% Glycine growth tolerant resembling CFF) could be isolated from heifers with infertility that were originally inseminated with semen from bulls known to have low fertility problems (*Park et al., 1962*). Furthermore, it was reported that serovar B strains of CFF could cause bovine infertility in about 5% of observed infections caused by *C. fetus* ssp. (*Dekeyser, 1984*). This type of contradictory information questions the established division in pathogenicity of infertility between CFF and CFV/CFVi as described above by the OIE and indicates that further research is needed in this regard.

One other problem in resolving issues between the subspecies is identification. Currently there are only a few phenotypic tests to differentiate CFF and CFV in most diagnostic laboratories. These include evaluation of growth on agar containing 1% glycine, generation of $H_2S$ from cysteine rich media, and growth at 42 °C (*OIE, 2018*). However these tests are poorly standardised, limited in number and complicated by CFVi strains, which are intermediate in their phenotype between CFF and CFV (*Van Bergen et al., 2008*) particularly with respect to cysteine metabolism, a feature which appears to be conferred by an L-cysteine transporter (*Farace et al., 2019*; *Van der Graaf-van Bloois et al., 2016*). One phenotypic characteristic, serotype analysis based on heat stable LPS antigens, has been fully characterized. Due to variation in both the genes (*sapA/sapB*) encoding surface array proteins and the surface lipopolysaccharide structure, CFF can be of serogroup A or B or AB (*Dworkin, Tummuru & Blaser, 1995*) but CFV is always serogroup A. However, this has limited use since CFF strains are mostly serotype A.

Overall, identification between subspecies using phenotypic and serological testing lacks either accuracy and/or reproducibility and, consequently, has equivocal usefulness.

Genetic based tools have been investigated in an effort to overcome this problem and to help correlate phenotypic and genomic characteristics of *C. fetus* subspecies (*OIE, 2018*; *Van der Graaf-van Bloois et al., 2014*). At the genomic level both pulse-field gel electrophoresis (PFGE) and amplified fragment length polymorphism (AFLP) have been reported to be useful methods for discriminating CFF and CFV (*On & Harrington, 2001*; *Wagenaar et al., 2001*) but the technicalities of these methods render them unsuitable for routine use in many laboratories. Multi-locus sequence typing (MLST), in which amplicons of seven housekeeping genes were compared at the sequence level, identified several distinct types but overall these studies highlighted the clonal nature of *C. fetus* and failed to correlate sequence type with subspecies and biovar identity (*Van Bergen et al., 2005*). Similarly rRNA gene loci were too highly conserved to be of value while sequences comprising the CRISPR-*cas* system, despite exhibiting significant variability, did not provide a useful means of *C. fetus* subspecies discrimination (*Calleros et al., 2017*). The suggestion that the two subspecies can be distinguished using a genomic island present only in CFV (*Gorkiewicz et al., 2010*) has not gained wide acceptance. Rapid and highly accurate methods involving amplification and detection of specific loci would be ideal if appropriate targets can be identified. To this end an early report targeted the carbon starvation gene (*cst*1) for *C. fetus* detection and the *par*A gene, carried either chromosomally or on a plasmid, for the identification of CFV (*Hum et al., 1997*). Attempts to adapt the latter assay to a real-time format have been described (*Chaban et al., 2012*; *McMillen et al., 2006*) despite reports demonstrating the lack of specificity of this target (*Schmidt, Venter & Picard, 2010*; *Spence et al., 2011*). Meanwhile, alternative targets that perform more consistently include the *nahE* gene that encodes a sodium-hydrogen exchanger family protein for detection of all *C. fetus* while the ISCfe1 transposase sequence is proposed as a specific marker for CFV, with the caveat that careful design of primers is needed for complete inclusivity of all samples carrying this sequence (*McGoldrick et al., 2013*; *Van der Graaf-van Bloois et al., 2013*). Genetic targets that specifically identify CFF have not yet been reported.

In an effort to better understand other aspects associated with CFF vs CFV, detailed comparisons of the genome organisation of CFF and CFV isolates have been undertaken using both draft and closed chromosome sequence information (*Ali et al., 2012*; *Kienesberger et al., 2014*; *Van der Graaf-van Bloois et al., 2014*). These studies concluded that all mammalian-associated CFs share extensive genome synteny and vary primarily due to the presence of distinct pathogenicity islands dotted throughout the chromosome. While these studies clearly indicated the high concordance of the genome complement in all isolates, the organisation of this genetic information can be revealed only through comparison of closed genomes. Unfortunately, the closed genome comparisons reported to date have typically involved only one sub-species member in each case, and thus lack consideration of the range of genomic diversity within each sub-species. Challenges in generating complete, polished and annotated genomes for these organisms are due to their

high AT content and large numbers of repetitive and mobile elements. As a result, most of the available genomic data for CF isolates is at the draft genome level only.

This report describes the use of multiple genomics methods and tools to generate closed genome sequences for seven phenotypically well characterised *C. fetus* isolates representative of the two mammalian-associated subspecies and their biovars. Comparison of these genomes with an additional seven complete and polished genomes recovered from the NCBI database illustrates that, with a few exceptions, all have a similar gene complement. However, their level of genome synteny varies enormously and mechanisms by which these changes may have occurred are explored.

These data add significantly to our understanding of the genomic variation within this bacterial species and should assist future studies to better understand its emergence and identify the specific features responsible for differences in tropism and pathogenicity. In addition, given the need to be able to accurately differentiate CFF from CFV/CFVi in cattle, these data may help, in practical terms, to identify useful additional markers for rapid and accurate subspeciation of isolates and provide a better understanding of the genes directly associated with BGC than currently exists.

## MATERIALS AND METHODS

### Isolate growth and phenotypic characterization

*Campylobacter fetus* ssp. isolates analysed in this study (Table 1) originated from clinical specimens from cattle submitted for culture. Five of the isolates were cultured in Canada and were identified originally to subspecies using a full array of tests as described elsewhere (*Devenish et al., 2005*). The two *C. fetus* strains from outside Canada were identified to subspecies/biovar in the submitting laboratory and a record maintained in the stock culture collection. Upon identification internally or receipt externally, pure culture isolates were inoculated into vials of CryoStor cryopreservative solution (Innovatek Medical Inc., Vancouver, BC, Canada) and frozen at $-80$ °C as part of a larger stock culture collection. For routine verification of identification in this study, the seven frozen *C. fetus* ssp. strains were cultured on Cysteine Heart blood (CH) agar or Mueller–Hinton agar containing 10% sheep blood and incubated under microaerophilic conditions (4% $O_2$, 9.5% $CO_2$, 86.5% $N_2$) for 2–4 days at 36 °C. A highly specific *C. fetus* monoclonal based ELISA procedure was used for genus, species and serotype (A or B) identification of the isolates (*Brooks et al., 2002*; *OIE, 2018*).

For subspecies and biovar identification of the seven isolates, all media were inoculated with a prepared McFarland standard 1.0 suspension of the isolate, incubated under microaerophilic conditions at 36 °C unless otherwise stated and tested for four different phenotypes as follows; (1) glycine tolerance—conducted in tubes of thioglycollate-135C broth supplemented with 0.0%, 0.6%, 1.0%, 1.3%, 1.5% and 1.9% glycine into which 0.1 ml of the bacterial suspension was inoculated. CFF isolates routinely grow at 1.3% glycine and above but never below, while CFV and CFVi isolates will grow at 1.0%, or sometimes weakly at 1.1%, and below but never at ≥1.3% glycine, (2) growth at 42 °C. A loopful of each isolate suspension was streaked onto two CH agar plates for well isolated colonies and incubated at 36 °C or 42 °C for 5 days. Using this procedure typically about 50% and

**Table 1 Characteristics of the 14 *C. fetus* isolates compared in this study.**

| Sample designation | Country of origin | Year of isolation | Source details | Serotype | Phenotypic identification |
|---|---|---|---|---|---|
| Samples analysed in this study | | | | | |
| CFF-02A725 | Canada—Alberta | 2002 | Bovine, Preputial wash | A | *C. fetus fetus* |
| CFF-09A980 | Canada—Quebec | 2009 | Bovine, Preputial wash | A | *C. fetus fetus* |
| CFF-00A031 | Canada—British Columbia | 2000 | Bovine, Preputial wash | A | *C. fetus fetus* |
| CFV-08A948 | Canada—Alberta (Premise 1) | 2008 | Bovine, Preputial wash | A | *C. fetus venerealis* |
| CFV-08A1102 | Canada—Alberta (Premise 2) | 2008 | Bovine, Preputial wash | A | *C. fetus venerealis* |
| CFVi-ADRI545 | Northern Australia | 1984 | Bovine, Reproductive tract | A | *C. fetus venerealis* biovar intermedius |
| CFVi-ADRI1362 | Argentina | 1989 | Bovine, Vaginal mucus | A | *C. fetus venerealis* biovar intermedius |
| Samples analysed in previous studies | | | | | |
| CFF-82-40 | USA | 1982 | Human | A | *C. fetus fetus* |
| CFF-04-554 | Argentina | 2004 | Bovine fetus | B | *C. fetus fetus* |
| CFF-NCTC10842 | France | unknown | Ovine, aborted fetus | B | *C. fetus fetus* |
| CFV-84-112 | USA | 1984 | Bovine | A | *C. fetus venerealis* |
| CFV-97-608 | Argentina | 1987 | Bovine | A | *C. fetus venerealis* |
| CFV-NCTC10354 | United Kingdom | 1952 | Bovine, heifer vaginal mucus | A | *C. fetus venerealis* |
| CFVi-03-293 | Argentina | 2003 | Bovine, aborted fetus lung | A | *C. fetus venerealis* biovar intermedius |

0.0% of CFF and CFV/CFVi isolates, respectively, show growth at the higher temperature of 42 °C, (3) $H_2S$ production-sensitive test. Brain heart infusion semi-solid medium with 0.022% L-cysteine was inoculated with 0.2 ml of the standard suspension followed by addition of a lead acetate strip. Incubation of tubes was for 5 days after which they were observed for blackening of the lead acetate strip. Traditionally, >95% of CFF and CFVi will be positive and show blackening of the lead acetate strip while 100% of CFV isolates will be negative, (4) ability to reduce selenite. A loopful of the bacterial strain immersed into sterile saline was streaked evenly over the surface of an Albimi agar slant containing 0.1% sodium selenite. The medium was incubated at 36 °C for 5 days and observed for selenite reduction (reddening in the medium). Consistently 100% of CFF and CFVi isolates show reduction of the selenite and while approximately 20% of CFV isolates can be positive in this test procedure the reduction is much weaker. Known positive and negative control bacterial strains were included for comparison in all testing procedures.

## DNA extraction

Colonies collected from agar plates were subjected to DNA extraction using a Wizard Genomic DNA extraction kit as per the supplier's directions (Promega, Madison, WI, USA) and quantification of the final solution was determined by a Qubit instrument using a broad sensitivity dsDNA quantification kit (Thermo Fisher Scientific, Waltham, MA, USA).

## Optical mapping

Each *C. fetus* isolate was grown on plates prepared with *Campylobacter* agar base (Oxoid CM0689; Thermo Fisher, Waltham, MA, USA) supplemented with 5% laked horse blood (Oxoid SR0048) under 5% $O_2$, 10% $CO_2$ at 37 °C for 2 days. Single colonies were collected during the exponential growth phase for preparation of high molecular weight genomic DNA using an Argus DNA Extraction Kit. DNA was loaded on a MapCard following the manufacturer's instructions (OpGen, Inc., Gaithersburg, MD, USA). After digestion with *NcoI* restriction enzyme, images of digested nucleotide molecules were captured and processed by an Argus Optical Mapping System (OpGen Inc., Gaithersburg, MD, USA) equipped with a fluorescent microscope. A restriction map of a closed, circular genome was generated using Argus MapSolver™ software (OpGen Inc., Gaithersburg, MD, USA). Each optical map was compared to the in silico *NcoI* map generated from the assembled DNA sequence produced by whole genome sequencing data of the same isolate.

## Whole genome sequencing

DNA extracts were subjected to multiple methods of whole genome sequencing. Pac Bio sequencing, performed by the McGill University and Genome Quebec Innovation Centre using PacBio RS II SMRT technology, generated between 45,000 to 107,000 raw reads, corresponding to 2 to $13 \times 10^8$ sequenced bases, per sample. Reads were error corrected and assembled using the Hierarchical Genome Assembly Process (HGAP) workflow (*Ihon, 2014*) to generate between 1 and 12 contigs per isolate with average reference genome coverage between 75 and 495. Contigs were ordered and oriented using the optical maps. The resulting draft genomes were used for reference based assemblies of paired end reads generated from indexed Nextera XT libraries run on an Illumina Hiseq 2000 sequencer ($2 \times 300$ except for CFF02A725 and CFV08A1102 which were run earlier using $2 \times 100$ reagent kits) at the Michael Smith Genome Sciences Centre (BC, Canada). Assemblies were initially generated and reviewed using the Lasergene Genomics suite software v14 (DNASTAR Inc., Madison, WI, USA) in an iterative process to fill gaps and remove small errors introduced by the PacBio sequencing; Illumina reads added >100 average base coverage. Small gaps in coverage for isolates CFF02A725 and CFV08A1102 were filled by Sanger sequencing of PCR products. Final assemblies were confirmed by comparison with the optical maps. To further confirm these genome assemblies and facilitate the assembly of any plasmids harbored by these isolates long read sequencing was performed on barcoded samples using a rapid sequencing kit run on a R9.4 MinION flow cell (Oxford Nanopore, Cambridge, MA, USA). The combined sequence data from all platforms (Illumina paired-end, Pac-Bio and Nanopore reads) were assembled using Unicycler (*Wick et al., 2017*). Briefly, this software employs the SPAdes tool for de novo assembly of the short-read data and then uses the long reads to bridge gaps in these assemblies. The bridged assemblies then underwent multiple rounds of short-read polishing.

## Annotation and genome interrogation

Chromosomal and plasmid sequences were annotated using Prokka (*Seemann, 2014*) with further refinement using an in-house script (*Duceppe, 2019*) that uses a clustering algorithm to improve predicted annotation descriptions. Due to variation in annotation designations for some genes the identification of comparable protein products in different isolates was, in some cases, performed by cross searching several isolates with the predicted protein products. CRISPR loci were identified using the web-based CRISPRFinder tool (*Grissa, Vergnaud & Pourcel, 2007*). All genome alignments were generated using the progressive Mauve option of Mauve version 2.3.1 (*Darling et al., 2004*) using previously characterised sequences as reference.

# RESULTS

## Phenotypic analysis

This study compares 14 representative mammalian-associated *C. fetus* isolates of which 12 originated from bovines while the remaining two CFFs are of human and ovine origin (Table 1). Seven of these isolates, which are described in this report for the first time, were all identified as *C. fetus* using a specific monoclonal antibody capture ELISA test and were found to possess serotype A heat stable lipopolysaccharide (LPS) antigens. Details of their provenance (Table 1) and the results of their phenotypic testing (Table S1) are presented. Based on the ability for growth on media with >1% glycine, generation of hydrogen sulphide in cysteine rich medium, the ability to reduce selenite and grow at 42 °C, three isolates scored as positive in either three or four tests and were identified as CFF. Two isolates which scored as negative by all four tests were classified as CFV while the remaining two isolates were identified as CFVi based on intermediate results that were shared with both CFV (inability to grow in broth with >1% glycine and at 42 °C) and CFF (production of $H_2S$ and reduction of selenite). However, it is evident from the results shown in Table S1 that a single biochemical test is sometimes insufficient to yield an accurate sub-type classification. Phenotypic analysis supporting the subspecies and biovar classification of the additional seven *C. fetus* isolates, as summarised in Table 1, has been reported previously (*Van der Graaf-van Bloois et al., 2014*).

## Genome sequencing and assembly

All seven newly characterised isolates were analysed by long (PacBio) and short (Illumina) read sequencing; all raw data, available in Genbank (Table S2), was used for generation of closed chromosomal sequences as described. To assist with and confirm the accuracy of these assemblies, each genomic sequence was used to generate a *Nco*I restriction map in silico for comparison with an experimentally generated optical map for each isolate (Fig. S1). These data indicate excellent concordance between both maps for all seven isolates, the non-aligned regions at the termini being a result of incomplete alignment of these circular chromosomes by a linear format. There was one small inconsistency in isolate CFF09A980 (Fig. S1) which could not be resolved, though the sizes of the fragments in this region of the genome did not appear to differ significantly between the optical map

**Table 2 Characteristics of the chromosomal genomes of the 14 *C. fetus* isolates compared in this study.**

| Isolate designation | Genome length (bp) | % GC content | No. of Genes[1] | No. of rRNA operons | No. of tRNA loci | NCBI Assembly Accession# |
|---|---|---|---|---|---|---|
| CFF-82-40 | 1,773,615[2] | 33.3 | 1,769 | 3 | 43 | CP000487 |
| CFF-02A725 | 1,782,312 | 33.3 | 1,872 | 3 | 44 | CP059974 |
| CFF-09A980 | 1,810,730 | 33.3 | 1,838 | 3 | 44 | CP059445 |
| CFF-00A031 | 1,777,789 | 33.3 | 1,797 | 3 | 44 | CP059443 |
| CFF-04-554 | 1,800,764[2] | 33.2 | 1,752 | 3 | 43 | CP008808 |
| CFF-NCTC10842 | 1,763,253 | 33.3 | 1,796 | 3 | 44 | LS483431 |
| CFV-84-112 | 1,926,886[2] | 33.3 | 1,992 | 3 | 43 | HG004426 |
| CFV-97-608 | 1,935,028[2] | 33.3 | 1,939 | 3 | 43 | CP008810 |
| CFV-NCTC10354 | 1,885,704 | 33.2 | 1,914 | 3 | 44 | CP043435 |
| CFV-08A948 | 1,956,863 | 33.3 | 2,046 | 3 | 44 | CP059441 |
| CFV-08A1102 | 1,956,865 | 33.3 | 2,048 | 3 | 44 | CP059439 |
| CFVi-03-293 | 1,866,009[2] | 33.3 | 1,821 | 3 | 43 | CP006999 |
| CFVi-ADRI545 | 2,019,872 | 33.4 | 2,156 | 3 | 44 | CP059437 |
| CFVi-ADRI1362 | 2,071,525 | 33.6 | 2,237 | 3 | 44 | CP059432 |

**Notes:**
[1] Including pseudogenes.
[2] Data recovered from *Van der Graaf-van Bloois et al. (2014)*.

and the sequence read data. The later addition of MinIon sequence data (Table S2) further confirmed these genome assemblies.

## Genome analysis

### Genome organisation

A summary of the characteristics of the closed circular chromosome of all 14 *C. fetus* isolates is presented in Table 2. Overall the genome size appeared to reflect the subspecies and biovar classification with all CFF strains having the smallest genomes, of approximately 1.8 Mb, while the genomes of the CFV and CFVi strains were larger, generally between 1.9 and 2 Mb with the exception of isolate CFVi03-293 which was a little smaller at ~1.87 Mb. GC content was consistent at 33.2–33.3% except for two CFVi isolates which had a slightly higher content.

Chromosomal alignments were generated to compare the overall organisation of these 14 CF genomes. Figure 1 shows a comparison of the overall genome organisation for three representative CFs, one each of CFF, CFV and CFVi while further comparisons between members of each subspecies and biovar are shown in Figs. 2–4. The alignment presented in Fig. 1 indicates an overall similar genome organisation for all three CFs but with two main observations: a hypervariable region, located variously between bases 450,000 to 600,000, following a well conserved block of sequence and increasing length of the genome from CFF to CFV and CFVi members due to multiple insertions.

The six CFF isolates (Fig. 2) exhibited a high level of synteny with significant differences limited to a few relatively small regions. The sequences corresponding to residues 435,000
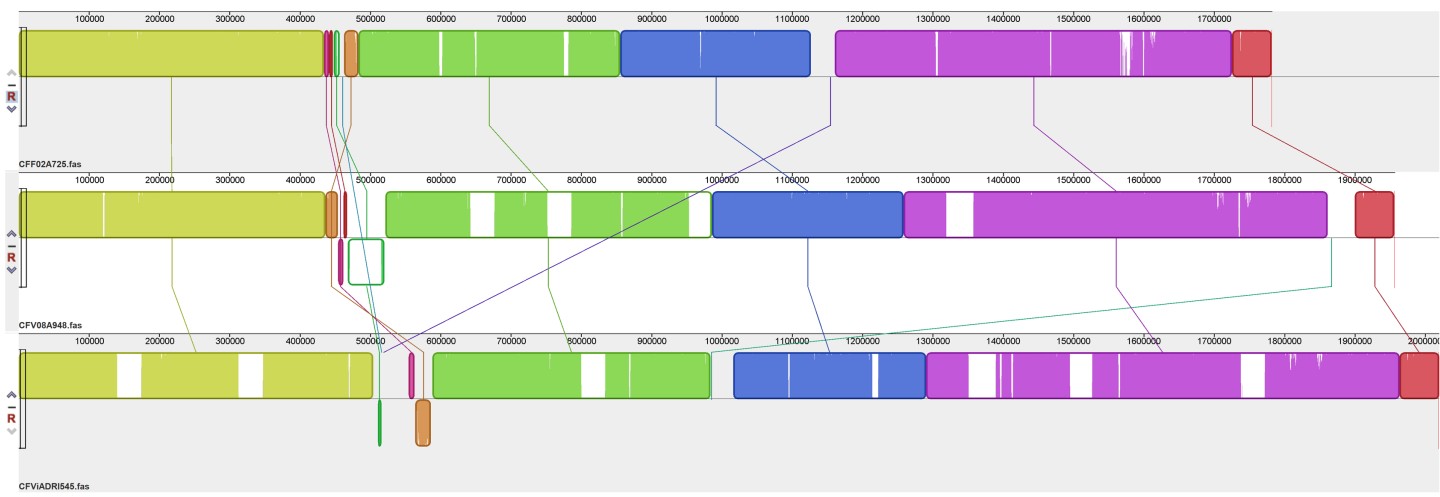

**Figure 1 Alignment of three CF genomes representative of CFF, CFV and CFVi.** Assembled genomes were aligned using the progressive Mauve option of Mauve software v2.3.1.

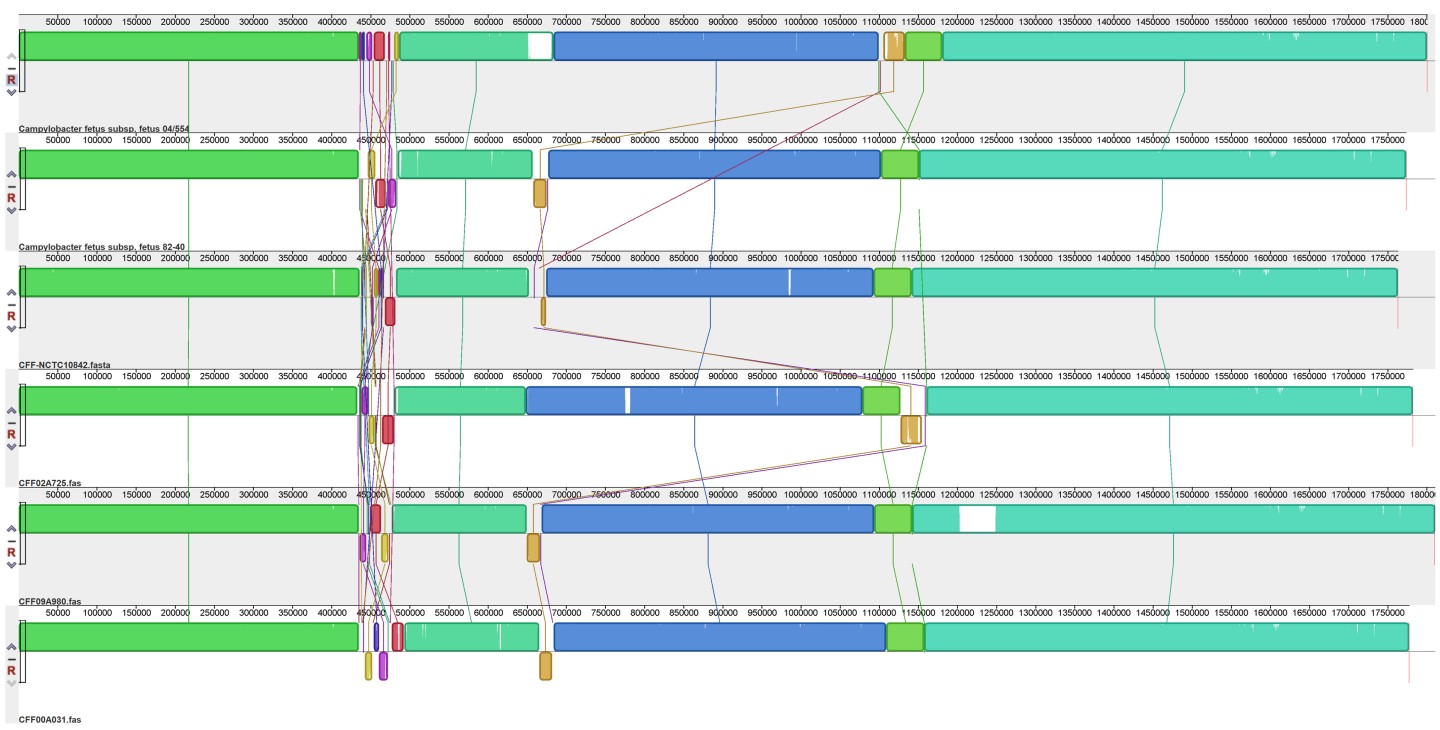

**Figure 2 Alignment of six CFF genomes.** Assembled genomes were aligned using the progressive Mauve option of Mauve software v2.3.1.

and 485,000 of the reference strain (CFF-04-554) were the most variable between all isolates and included significant rearrangements and gene inversions of blocks of sequence. There was also a variable sequence of ~32 kb (corresponding to the white/pale orange block at bases 1,100,000–1,113,200 of CFF04-554) which contained elements representative of a prophage that appeared in different locations in each isolate.
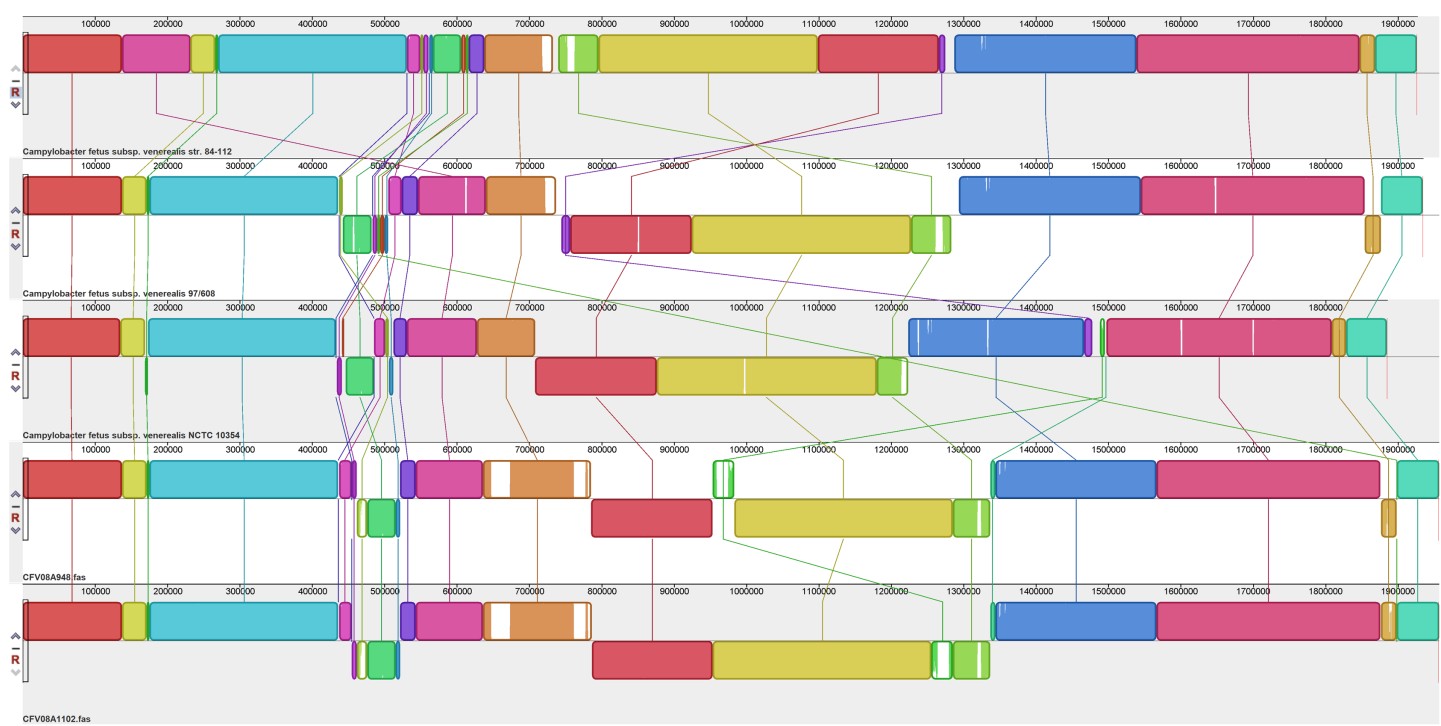

**Figure 3 Alignment of five CFV genomes.** Assembled genomes were aligned using the progressive Mauve option of Mauve software v2.3.1.

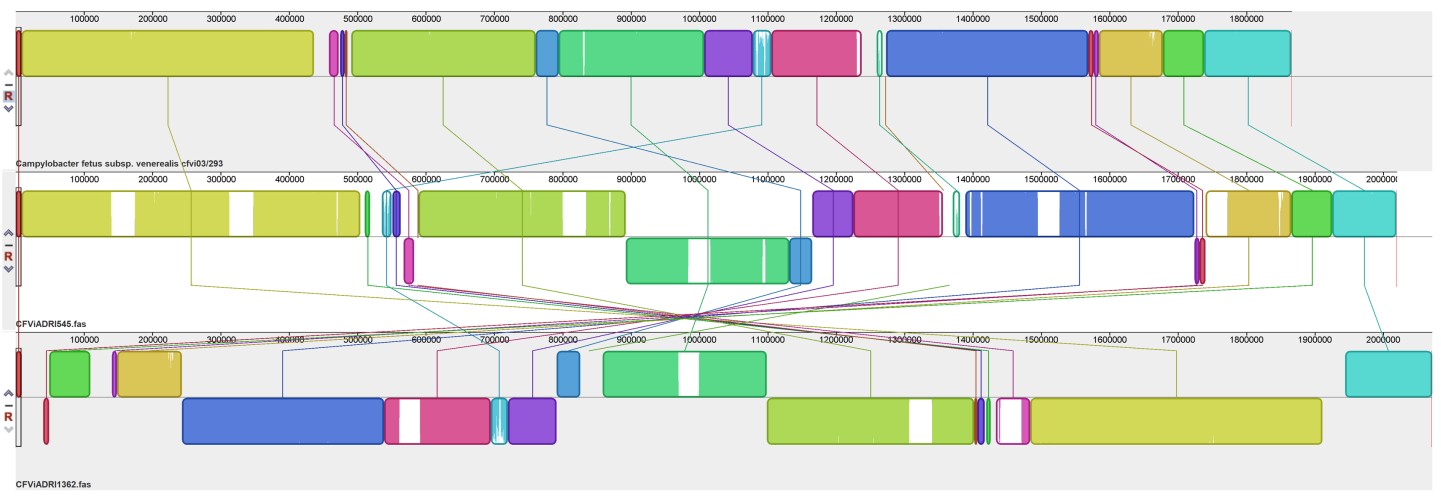

**Figure 4 Alignment of three CFVi genomes.** Assembled genomes were aligned using the progressive Mauve option of Mauve software v2.3.1.

Additional variations were observed in specific isolates including insertions of ~45.4 kb (1,203,385–1,248,755) in CFF-09A980 (described further below) and one of ~30 kb (650,000–680,000) in CFF 04-554 which encoded many hypothetical proteins and several different Fic protein alleles.
The alignment of five CFV genomes (Fig. 3) again identified a hypervariable region (corresponding to residues 532,000–616,675 of CFV-84-112) but it also revealed greater variability in other genomic regions amongst this group than was observed for the CFFs. Compared to the other CFVs, the CFV-84-112 genome included a transposition of a block of sequence of about 95 kb; the region affected (bases 142,200–237,000), which was bound by rRNA small and large subunit genes, corresponded to residues 544,240–638,200 of the other genomes. This genomic region encoded several membrane and periplasmic proteins, several of which had either signal transduction or transporter functions, including heavy metal transportation, products involved in heavy metal resistance and several enzymes involved in multiple metabolic pathways. A large genomic segment of CFV-84-112 corresponding to residues 761,700–1,275,130 was inverted compared to the other genomes. A 22 kb sequence, representative of a genomic island harbouring Type IV secretion system genes (see below), located close to the downstream terminus was inverted in both CFV-97-608 and CFV-08A948 compared to the other three CFVs.

Yet additional variability was evident amongst the three CFVi isolates (Fig. 4). Indeed, when, as per convention, the *dnaA* gene is placed at the start sequence, CFVi-ADRI1362 exhibited an inversion of a large section of the genome compared to the other CFVi isolates such that genome co-ordinates were significantly altered, though in reality this difference could be visualised as an inversion of a smaller genome segment of ~405,000 bp. CFVi-ADRI545 exhibited a genome organisation closer to that of the reference genome but with several notable differences. The hypervariable region of this isolate was shifted downstream due to two insertions; indeed, this sample contained no less than six insertions relative to the reference as well as an inversion of residues 894,000–1,165,000 compared to the reference, a feature also shared by CFVi-ADRI1362. The features contributing to these differences are detailed further below.

### Hypervariable region

Annotation information for all 14 genomes facilitated comparison of their gene complement including a detailed comparison of the hypervariable region described above and recognised previously as a locus encompassing multiple *sap* gene alleles (*Tu et al., 2003*). Comparison of the organisation of this region is shown in Fig. 5 for 13 CFs, CFV-08A948 and CFV-08A1102 being virtually identical. This region includes two sets of gene groupings retained in virtually all isolates but varied in order and orientation. One set includes the genes *tolC* (an outer membrane protein), *prsE* and *prsD*, which encode products involved in the Type 1 secretion system, and a presumed peptidase. Notably this gene group was absent in the single ovine isolate CFF-NCTC10842 but it is unknown if this is typical of ovine isolates. The second set of 11 genes, present in all cases, is bound by the *ssrA* gene encoding a tmRNA and the *ybeY* gene that encodes an endoribonuclease, except for CFV-84-112 in which the *ssrA* gene is missing. Between these two loci are genes encoding three transferases, a rhodanese, the acid membrane antigen A and four genes (*mlaB, mlaD, mlaE* and *MlaF*) all believed to be involved in lipid transport. In CFF-82-40 the *mlaE* locus is identified as a pseudo gene. Interspersed around these two gene groupings are multiple copies of the surface array protein genes, *sapA* for all 12 serogroup

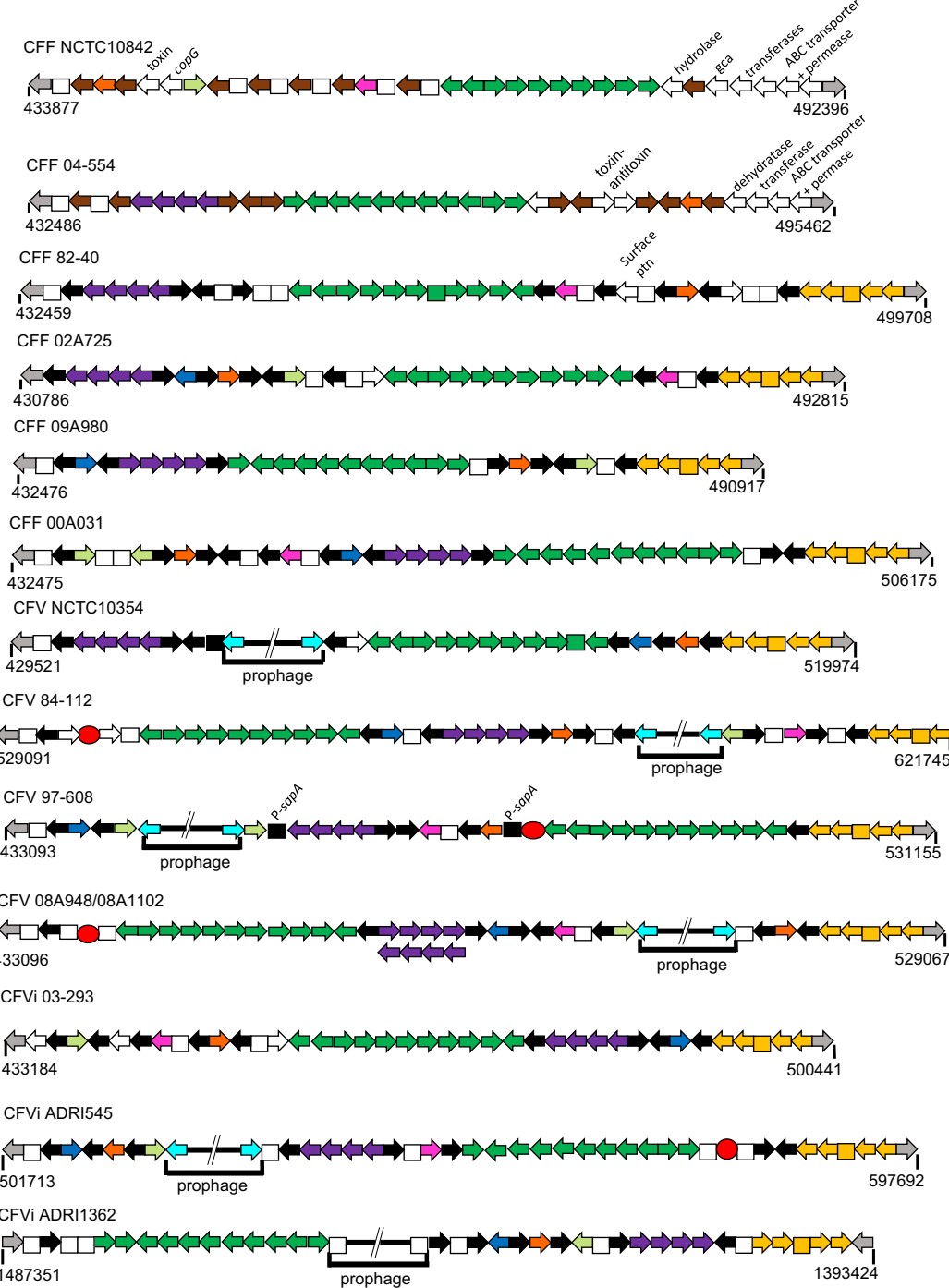

**Figure 5 Schematic of the hypervariable region for 14 CF isolates.** Arrows, which are not drawn to scale, indicate the gene orientation. Squares indicate either hypothetical or pseudo genes. Grey arrows at the termini indicate genes common to all isolates, that is, an upstream diheme cytochrome c and a downstream glycosyl transferase (*gtf*). Other colour codes represent genes thus: black, *sapA*; brown, *sapB*; emerald green, an 11 gene cassette comprising *ssrA* (which encodes a tmRNA), a histidine phosphotransferase, *panB*, a sulfur transferase, *ruvB*, *amaA*, *mlaE*, *mlaF*, *mlaD*, *mlaB* and *ybeY* (an endoribonuclease); purple, a four gene cassette comprising *tolC*, *prsE*, *prsD* and a peptidase; yellow, a five gene set comprising *gtf*, *mtfA*, a hypothetical product and ABC transporter and permease genes; pink, *yafQ*; orange, AAA family ATPase; blue, *hipA*; light green, surface protein; white (unlabelled), transporters of
**Figure 5 (continued)**
variable sequence; turquoise, *lexA*, a repressor at the boundary of most prophages; additional novel genes are indicated by white arrows and labelled individually. Red circles identify IS605 elements. The numbers below each schematic indicate the start and stop locations of the terminal gene ORFs. Note that the CFVi-ADRI1362 schematic is shown in reverse orientation relative to the others. Samples CFV-08A948 and CFV-08A1102 were identical except for the inversion of the purple cassette in the latter.

A isolates and *sapB* for the two serogroup B isolates, CFF-04-554 and CFF-NCTC10842. Copy numbers of the *sap* genes range from six to 10. Additional genes found in this region for some isolates encode a mRNA interferase toxin (*yafQ* also known as *relA*), a hip A domain protein *(hipA)*, an AAA family ATPase, additional surface associated products (sap) and some of unknown function. Indeed, the high proportion of genes that encode proteins involved in functions at the bacterial surface indicate the important pathogenic nature of this hypervariable region for all CFs.

### Selected gene complement

Many of the genes of the hypervariable region, together with several additional genes presumed to be important with respect to virulence or phenotype, are detailed in Table S3. In most cases these genes have been identified in all members of the *Campylobacter* genus (*Ali et al., 2012*) and are presumably necessary to support host cell adherence, invasion and immune evasion. As such in general they were retained in all 14 CFs though with significant variation in location for some isolates. This included three copies of the cytolethal distending toxin operon comprised of three genes (*cdtA, cdtB* and *cdtC*), with truncation of one or more alleles in some cases. In some isolates specific genes were lacking, including loci encoding a tyrosine recombinase (*xerH*), a filamentous hemagglutinin transporter protein (*fhaC*) or the twitching mobility protein (*pilT*) but these variations did not respect the sub-species/biovar designations.

However, other coding differences did respect some of the observed phenotypic variation. All five CFV isolates lacked two genes (*tcyB* and *tcyC*) comprising the L-cys ABC transporter operon while the 3′-terminal sequence of the remaining gene (*tcyA*) was modified; all CFF and CFVi isolates retained the complete unaltered operon. Three genes previously described as significant with respect to CF variation in lipopolysaccharide biosynthesis (*Kienesberger et al., 2014*) were also examined. The *glf* locus that encodes a UDP-galactopyranose mutase was found in all serotype A CFFs but not in serotype B CFFs or any CFV/CFVi isolates. In contrast the *mat1* gene that encodes a maltose O-acetyltransferase activity was present in all isolates except for the serotype A CFFs. A *wcbK* gene that encodes the enzyme GDP-mannose 4,6 dehydratase was found only in serotype B CFFs. Finally, it was noted that the transporter gene (annotated as *kefC* in this study) corresponding to the *nahE* target used previously for *C. fetus* detection was present in all isolates.

### Mobile genetic elements

The presence of two groups of mobile genetic elements, transposons and prophages, throughout these 14 CF genomes was highly variable (Table 3). Various copy numbers of

**Table 3 Summary of transposons and prophages identified in *C. fetus* genomes**

| | IS605 element | | IS607 element | | Prophage sequence | |
|---|---|---|---|---|---|---|
| | Copy number | Location | Copy number | Location | Copy number | Location of phage portal protein |
| CFF-82-40 | 0 | | 0 | | 1 | 660,498–661,559 |
| CFF-02A725 | 0 | | 0 | | 1 | 1,140,631–1,140,831**; 1,141,256–1,141,690** |
| CFF-09A980 | 0 | | 0 | | 1 | 651,841–652,902 |
| CFF-00A031 | 0 | | 0 | | 1 | 667,640–668,701 |
| CFF-04-554 | 0 | | 0 | | 1 | 1,119,046–1,120,107 |
| CFF-NCTC10842 | 0 | | 0 | | 1 | 670,040–671,101 |
| CFV-84-112 | 2 | 533,924–535,307; 1,868,521–1,870,054 | 0 | | 3 | 585,408–586,472; 739,112–740,185; 1,277,904–1,278,977 |
| CFV-97-608 | 3 | 507,749–509,438; 850,118–851,810; 1,876,554–1,878,246 | 6 | 120,302–122,369; 456,444–458,511; 611,586–613,653; 827,189–829,256; 1,647,029–1,649,096; 1,869,088–1,871,155 | 3 | 464,286–465,362; 743,287–744,360; 1,283,953–1,285,026 |
| CFV-NCTC10354 | 3 | 1,236,767–1,238,459 1,332,720–1,334,412 1,699,299–1,700,991 | 3 | 779,640–781,707 996,739–998,806 1,599,993–1,602,060 | 1 | 466,844–467,920 |
| CFV-08A948 | 5 | 439,037–440,570; 1,343,995–1,345,528; 1,734,964–1,736,497; 1,874,884–1,876,417; 1,898,499–1,900,032 | 3 | 120,406–122,323; 857,035–858,952; 1,890,970–1,892,887 | 4 | 496,550–497,626; 654,579–655,652; 773,210–774,283; 965,366–966,439 |
| CFV-08A1102 | 5 | 439,037–440,570; 1,343,999–1,345,532; 1,734,965–1,736,498; 1,874,886–1,876,419; 1,898,501–1,900,034 | 3 | 120,406–122,323; 857,221–859,138; 1,882,031–1,883,948 | 4 | 496,551–497,627; 654,580–654,792** + 655,217–655,654** 773,214–774,287; 1,272,652–1,273,725 |
| CFVi-03-293 | 3 | 1,006,929–1,008,846; 1,094,859–1,096,392; 1,249,187–1,250,720 | 3 | 420,725–422,642; 758,891–760,808; 1,083,526–1,085,433* | 1 | 1,088,131–1,089,192 |
| CFVi-ADRI545 | 4 | 580,856–582,389; 868,430–869,963; 1,094,766–1,096,299; 1,358,484–1,360,018 | 4 | 1,395,744–1,397,661; 1,411,977–1,413,894; 1,564,281–1,566,198; 1,738,304–1,740,221 | 7 | 161,401–162,462; 325,552–326,613; 536,549–537,625; 821,342–822,403; 996,953–998,014; 1,513,357–1,514,418; 1,752,790–1,753,851 |
| CFVi-ADRI1362 | 0 | | 5 | 47,649–49,566; 707,540–709,457; 790,038–791,955; 1,098,560–1,100,477; 1,498,399–1,500,316 | 9 | 20,800–21,861; 128,294–129,355; 573,651–574,712; 711,744–712,805; 838,573–839,634; 981,695–982,768; 1,320,740–1,321,801; 1,452,009–1,453,070; 1,930,918–1,931,979 |

**Notes:**
* Transposase reported as pseudogene.
** Truncated product.

two distinct versions of the IS200 family of insertion sequences, IS605 and IS607, were scattered throughout the genomes of all CFV and CFVi isolates and were frequently present in the hypervariable region. Indeed, while most of these isolates contained multiple

**Table 4 Summary of the CRISPR-*cas* loci present in 14 *C. fetus* isolates.**

| Isolate | 1 | 2 | 3 | 4 | cas1 | cas2 | cas3 | cas4 | cas5 | cas6 | cas6_2 | cas9 | cas10 | RAMP family protein | Type III RAMP protein | Type III protein | CRISPR-associated protein |
|---|---|---|---|---|---|---|---|---|---|---|---|---|---|---|---|---|---|
| | CRISPR locus | | | | Locus ID of CRISPR-associated genes | | | | | | | | | | | | |
| | Spacer Number | | | | | | | | | | | | | | | | |
| CFF-82-40 | 21 | 26 | | | 665 | 666 | 663 | 664 | 662 | 659 | 1451 | | 1674 | | 1677 | 1678 | 1679 |
| CFF-02A725 | 2 | | | | | | | | | | 7255 | 7100 (P) | 8360** 8365** | 8370 (P) | 8380 (P) | 8385 | 8390 |
| CFF-09A980 | 23 | 24 | | | 3290 | 3295 | 3280* | 3285 | 3275 | 3260 | 7335 | 7185 (P) | 8435 | 8440 | 8450 | 8455 | 8460 |
| CFF-00A031 | 23 | 21 | | | 3330 | 3335 | 3320* | 3325 | 3315 | 3300 | 7130 | 6980 (P) | 8225 | 8230 | 8240 | 8245 | 8250 |
| CFF-04-554 | 5 | | | | | | | | | | 1463 | | 1676 | 1677 | 1679** | 1680 | 1681 |
| CFF-NCTC10842 | 8 | 38 | 5 | 6 | 661 | 662 | 659 | 660 | 658 | 655 | 1427 | 1396 (P) | 1649 | 1650 | 1652 | 1653 | 1654 |
| CFV-84-112 | 24 | | | | | | | | | | 15910 | | 18080 | 18090 | 18110 | 18120 | 18130 |
| CFV-97-608 | 25 | | | | | | | | | | 1589 | | 1801 | 1802 | 1804 | 1805 | 1806 |
| CFV-NCTC10354 | 23 | | | | | | | | | | 1525 | | 1738 | 1739 | 1741 | 1742 | 1743 |
| CFV-08A948 | 25 | | | | | | | | | | 8090 | 7935 (P) | 9190 | 9195 | 9205 | 9210 | 9215 |
| CFV-08A1102 | 25 | | | | | | | | | | 8080 | 7925 (P) | 9180 | 9185 | 9195 | 9200 | 9205 |
| CFVi-03-293 | 3 | | | | | | | | | | 7500 | | 8560 | 8565 | 8575 | 8580 | 8585 |
| CFVi-ADRI545 | 23 | | | | | | | | | | 8590 | 8440 (P) | 9895 | 9900 | 9910**–9920** | 9925 | 9930 |
| CFVi-ADRI1362 | 20 | | | | | | | | | | 1815 | 1965 (P) | 455 | 460 | 470 | 475 | 480 |

**Notes:**
* Sequence extended at 5′ end.
** Truncated sequence.
(P), pseudogene.

copies of both types of transposon, CFV-84-112 contained two IS605 copies only and CFVi-ADRI1362 contained five copies of IS607 but no IS605. These elements were not present in any of the CFF genomes.

All CFs contained a prophage sequence ranging in length between 30 and 35 kb but whereas the CFFs contained just one copy of this element all of the CFV and CFVi genomes, with the exception of CFV-NCTC10354 and CFVi-03-293, contained multiple copies; CFVi-ADRI1362 had no less than nine copies. This element was typically bound at either end by a copy of the *lexA* gene that encodes a prophage regulator, except for the prophage located in the hypervariable region of CFVi-ADRI1362 in which these loci were absent. The prophage sequence typically included genes encoding putative glutamine ABC transporter permease proteins, a mobile element, a modification methylase DpnIIA (*dpnM*), capsid and tape measure domain-containing proteins, an ATPase, phage and portal proteins, the GP27 locus and many hypothetical products.

### *CRISPR-cas complement*

The highly variable number of mobile elements present in these genomes led to a review of the presence of genes known to be involved in preventing invasion of the cell by foreign nucleic acids. This included a review of the components of the CRISPR-*cas* system.
All 14 isolates contained at least one CRISPR locus and several of the CFFs contained two or more such loci (Table 4). Each locus contained 30 base direct repeats (DRs) of sequence

GTTTGCTAATGACAATGTTTGTGTTGAAAC with occasional minor modification. Three isolates contained just one short locus of two (CFF-02A725), three (CFVi-03-293) and five (CFF-04-554) spacers respectively while the other isolates retained loci comprising 20–26 spacers. There was also significant variation in the presence of CRISPR-associated proteins. The four CFFs having two or more CRISPR loci encoded a complete set of *cas* genes (*cas1* to *cas6*) while the other two CFFs retained a modified *cas6* gene only. All seven CFV/CFVi isolates also contained this modified *cas*6 gene but lacked *cas*1-5 genes. A *cas9* pseudogene was identified in seven isolates representative of all three biotypes while this locus was not observed in the remaining isolates. With a few exceptions most other CRISPR-associated genes were retained though some had truncated ORFs.

### Restriction-modification system genes

Review of the presence of restriction-modification (R-M) system genes, also capable of limiting invasion of the cell by foreign genetic material, indicated that most isolates retained a common but limited set of such genes though the serotype B CFFs were somewhat distinct in this regard due to a lack of several Type 1 R-M loci (Table S3). However, review of the complement of so-called orphan methyltransferases that are unassociated with restriction endonucleases revealed some interesting differences. While all isolates retained one copy of an adenine-specific methyltransferase (*fokI*) all but one (CFF-04-554) contained one or more copies of a DNA adenine methylase (*dam*) gene. This gene appeared to be present as three distinct alleles which differ at their 5′ termini. Sequence annotations predict that allele 1 encodes a product of either 163 or 213 residues, depending on the start codon employed, while alleles 2 and 3 encode products of 253 and 270 amino acids respectively. Allele 1 was found only in CFF strains; allele 2 only was present as a single copy in CFF-02A725 and as multiple copies in most of the CFV/CFVi isolates. Allele 3 was restricted to five CFV/CFVi strains.

### Type IV secretion system genes

In light of the significance made of the conservation of Type IV secretion system (Type IV SS) genes previously suggested for CFV isolates (*Gorkiewicz et al., 2010*), the presence of such genes in all CFs was investigated. Several distinct gene cassettes containing different combinations of genes of this group were identified (Table 5) and the structure of these genomic islands is illustrated (Fig. 6). A genomic island (T4SS GI 1) of just under 40 kb containing several Type IV SS genes, including *virB2* to *virB11* and *virD4*, as well as genes believed to provide ancillary functions (Fic proteins and a lytic transglycosylase) and genes frequently plasmid-associated, such as *dnaG, topB* and a tetracycline resistance ribosomal protection protein Tet(44), were identified in CFF-09A980. This sequence, corresponding to the insertion identified between 1,200,000 and 1,250.000 in this isolate, was not present in any other CFF. A GI similar in composition and organisation was also found in CFV-84-112 and CFV-97-608 while CFVi03-293 and CFVi-ADRI545 retained a modified version of this GI in which some of the downstream genes were lost. Additional GIs harbouring a more limited group of these genes were also found. CFV-97-608 and CFV-84-112 harboured T4SS GIs 2 and 3

**Table 5 Type IV Secretion System Genomic islands in 14 *C. fetus* isolates.**

| | T4SS GI 1 | | T4SS GI 2 | | T4SS GI 3 | | T4SS GI 4 | | T4SS GI 5 | |
|---|---|---|---|---|---|---|---|---|---|---|
| | Copy number | Location | Copy number | Location | Copy number | Location | Copy number | Location | Copy number | Location |
| CFF-82-40 | 0 | | | | | | | | | |
| CFF-02A725 | 0 | | | | | | | | | |
| CFF-09A980 | 1 | 1,203,385–1,252,250 | | | | | | | | |
| CFF-00A031 | 0 | | | | | | | | | |
| CFF-04-554 | 0 | | | | | | | | | |
| CFF-NCTC10842 | 0 | | | | | | | | | |
| CFV-84-112 | 1 | 1,300,266–1,330,033 | | | 1 | 1,845,456–1,854,063 | | | | |
| CFV-97-608 | 1 | 1,306,338–1,336,107 | 1 | 1,841,346–1,878,197 | | | | | | |
| CFV-NCTC10354 | 1 | 1,224,281–1,255,740 | | | | | | | | |
| CFV-08A948 | 0 | | | | | | 1 | 1,331,503–1,357,485 | | |
| CFV-08A1102 | 0 | | | | | | 1 | 1,331,507–1,357,489 | | |
| CFVi-03-293 | 1 | 1,250,766–1,272,991* | | | | | | | 1 | 1,788,240–1,806,336 |
| CFVi-ADRI545 | 1 | 1,358,484–1,389,267** | | | | | | | | |
| CFVi-ADRI1362 | 0 | | | | | | | | 1 | 1,993,754–2,011,850 |

**Notes:**
* Integrase to lytic transglycosylase only.
** Integrase to *dnaG* only with highly modified product sequences.

respectively while CFV-08A948 and CFV-08A1102 harboured only T4SS GI 4 which encompassed most core Type IV SS genes, *dnaG* as well as a DNA topoisomerase. Notably T4SS GIs 2 to 4 each incorporated an IS element likely explaining their transposition and modification. CFVi-ADRI1362 contained only the T4SS GI 5 which harboured just four Type IV SS genes of highly modified sequence together with a topoisomerase, *EcoRI* methylase and *mobC* genes.

## Plasmid sequences and annotation

The hybrid assembly process, which also included MinION sequence data, permitted the identification and assembly of plasmid-like elements found in six of the seven isolates characterised in this study. Of the complete collection of 14 isolates, three CFF strains appeared to lack these elements, the other three CFFs each contained a single plasmid and the CFV and CFVi isolates harboured between one to four plasmids each (Table 6). Annotation summaries of these plasmids are provided (Table S4).

The genetic complement of these plasmids generally included multiple genes encoding conjugal transfer proteins of the Tra and Trb families that are involved in plasmid

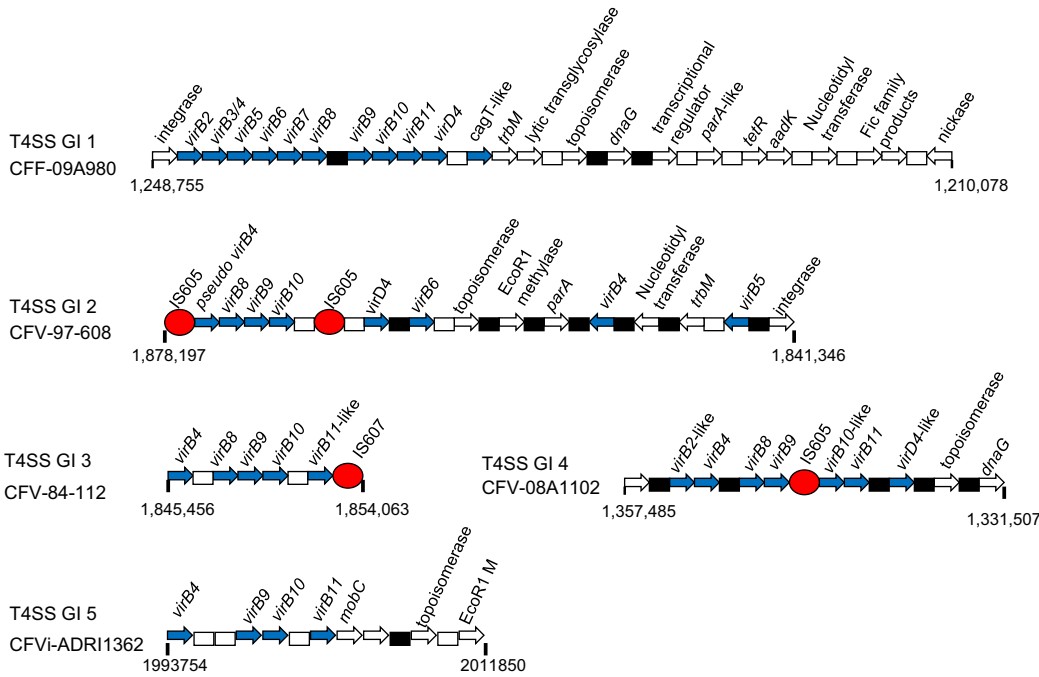

**Figure 6 Schematics of representatives of the five types of T4SS genomic island found in CFs.** Arrows indicate the gene orientation; hypothetical genes are represented by boxes, white for singles and black for multiple. Blue arrows indicate Type IV SS members though in some cases there was significant variation in the primary sequence of some alleles. Other genes shown in white are frequently plasmid-associated. Red circles indicate IS605 elements. The range of each GI is indicated below each schematic.

replication and in most cases several genes of the Type IV secretion system. Except for CFVi-03-293, at least one copy of the *parA* gene (designated as *soj* in our annotations), which encodes a plasmid partitioning protein, was found in all plasmid-harboring isolates, including the CFFs. This observation reinforces prior findings that this gene is not a specific marker for CFV/CFVi strains. Genes encoding various toxin-antitoxin products (*yafQ, vapC*) and a phage repressor protein of the BRO family were present on plasmids in all 11 isolates except for CFF-04-554 (yafQ, BRO), CFV-08A948 (BRO) and CFVi-03-293 (BRO). CFV and CFVi isolates harboured additional genes including Fic proteins (except CFV-84-112), relaxases and helicases, lytic transglycosylases, transcriptional regulators, as well as IS605 and IS607 elements which were often present in multiple copies. The plasmids of CFVi-ADRI1362 encoded several additional products that appear to have been obtained from a variety of different proteobacteria, including additional toxins, various enzymatic activities, ABC transporter components, a phage anti-repressor, the cag pathogenicity island protein, cpp46 and copg family domain proteins. The P3 plasmid of CFVi-ADRI1362 and the P2 plasmid of CFVi-03-293 encoded essentially the same products, including a Type IV SS operon, virtually complete except for a short internal segment. Both strains also contained the same cryptic plasmid of 3,993 bp despite some differences in the predicted coding capacity of this sequence. The remaining plasmids were however quite distinct. CFVi-ADRI1362-P1 harboured several copies of IS605, an element which is absent from the chromosome of this isolate

**Table 6 Listing of the plasmids found in 11 of 14 *C. fetus* isolates.**

| Isolate | Plasmid, size (bp) | Number of genes | NCBI Accession number |
|---|---|---|---|
| CFF-82-40 | | | |
| CFF-02A725 | | | |
| CFF-09A980 | P1, 52,345 | 66 | CP059446 |
| CFF-00A031 | P1, 26,793 | 36 | CP059444 |
| CFF-04-554 | P1, 25,862 | 32 | CP008809 |
| CFF-NCTC10842 | | | |
| CFV-84-112 | P1, 61,142 | 73 | HG004427 |
| CFV-97-608 | P1, 38,272 | 43 | CP008811 |
| | P2, 27,124 | 35 | CP008812 |
| CFV-NCTC10354 | P1, 27,915 | 36 | CP043436 |
| CFV-08A948 | P1, 38,770 | 50 | CP059442 |
| CFV-08A1102 | P1, 37,205 | 49 | CP059440 |
| CFVi-03-293 | P1, 91,400 | 124 | CP007000 |
| | P2, 35,326 | 38 | CP007001 |
| | P3, 3,993 | 6 | CP007002 |
| CFVi-ADRI545 | P1, 48,693 | 68 | CP059438 |
| CFVi-ADRI1362 | P1, 40,588 | 58 | CP059433 |
| | P2, 36,566 | 36 | CP059434 |
| | P3, 35,640 | 38 | CP059435 |
| | P4, 3,993 | 5 | CP059436 |

Note:
  Annotations of all plasmids are provided in Table S4.

(Table 3). The high genetic plasticity provided by these extrachromosomal elements was especially evident for the two CFV isolates, CFV-08A948 and CFV-08A1102, both collected in the province of Alberta, Canada, in 2008 from separate AI facilities. The chromosomes of these two isolates were highly clonal thus suggesting a common source of infection. However, their plasmids did exhibit some coding differences. CFV-08A948P1 contained two copies of a rha family transcriptional regulator and a *trbE* gene not found in CFV-08A1102P1 as well as one additional IS605 transposon. Products encoded by CFV-08A1102P1 but not CFV-08A948P1 included a BRO phage repressor protein, a single stranded binding protein and an adenosine monophosphate protein transferase (*vbhT*). In addition to these differences some gene products, including those of *trbF, trbL* and *trbJ*, exhibited significant sequence differences.

## DISCUSSION

Many studies have identified the clonal nature of *C. fetus* isolates (Van Bergen et al., 2005), a fact that has prompted studies on these organisms to better understand the genetic basis for their variable phenotype and host tropism. Phylogenetic analysis of MLST data has revealed the evolution of *C. fetus*, beginning with the emergence of distinct lineages normally associated with reptilian and mammalian hosts respectively, followed by division of the mammalian lineage into two serotypes, A and B, and the later emergence of the CFV subspecies from the A serotype (Dingle et al., 2010; Wang et al., 2013).

The studies presented in this report and by others (*Kienesberger et al., 2014*; *Van der Graaf-van Bloois et al., 2016*) strongly suggest that the recent emergence of the *C. fetus venerealis* spp. occurred in parallel with increased genome plasticity associated with the acquisition, through horizontal gene transmission (HGT), of multiple mobile elements including members of the IS200 family of transposons, prophages and plasmid-like extrachromosomal elements. Moreover, the greater genome plasticity exhibited by the CFVi isolates does suggest that these isolates represent a distinct group.

While differences in the presence or absence of these elements can be scored using draft genome sequence data, their context can be determined only through the examination of closed, polished genomes. Relatively few studies on *C. fetus* genomic structure have used such data for subspecies comparison and those reports have generally included very few (often just two) isolates (*Ali et al., 2012*; *Kienesberger et al., 2014*; *Van der Graaf-van Bloois et al., 2014*, *2016*). This report, which is the first to compare the organisation and gene complement of a significant number of complete closed *C. fetus* genomes, provides additional insights into the role that different elements may have played in the evolution of these genomes, particularly with respect to the emergence of CFV and CFVi.

Of the two types of mobile elements found in the CF chromosome, only CFV/CFVi strains harbored members of the IS200 transposon family, often in multiple copies as previously reported (*Van der Graaf-van Bloois et al., 2013*), and thus the IS*Cfe*1 element has become a target for development of sub-species discriminatory PCR-based tools (*Abril et al., 2007*; *McGoldrick et al., 2013*). Clearly the spread of these elements throughout the genome raises the possibility of gene disruption and function loss though it is notable that many of these elements were not located at points of genome rearrangement and thus may not be a primary driver of large-scale genome plasticity.

In contrast all CF isolates examined in this report contained prophage sequence. While the CFFs contained just one copy of this element, CFV/CFVi strains tended to harbour greater copy numbers of these elements and very notably these prophage sequences were often located at the boundaries of sequence rearrangements and inversions supporting the importance of this element type to genome plasticity. The potential role of prophage mobility in altering the gene complement is clearly illustrated by the sequence of CFF-02A725 in which prophage translocation (residues 1,127,063–1,160,613) resulted in the loss of several components of the CRISPR-*cas* system. The presence of prophage sequence in the hypervariable region of CFV and CFVi isolates may well have contributed to its evolution though its absence in that region of the CFFs suggests that other factors, including possibly homologous recombination involving the *sap* gene loci, may have contributed to the plasticity of this region (*Tu et al., 2003*). Indeed, the identification of rare recombinants having an AB serotype supports the high plasticity of this genomic region (*Dingle et al., 2010*).

Bacteria have a number of mechanisms to limit the extent to which foreign genetic material can invade a cell and become incorporated into the chromosome. Restriction-modification systems of many bacteria facilitate degradation of invading sequences having specific sequence motifs; however, *C. fetus* appears to have limited

capability in this regard with no clear distinction detected between CFF and CFV subspecies. Another important process involves the CRISPR-*cas* system that is believed to confer an adaptive immune response to protect against invasion by mobile genetic elements (*Van der Oost et al., 2014*). The CRISPR locus consists of a series of repeat sequences interspersed with spacer sequences derived from invading genetic material. Cas proteins are responsible for acquisition of these spacers as well as processing of RNA transcripts of these loci in a process that ultimately results in the degradation of DNA homologous to the spacer sequence. Three distinct CRISPR-*cas* systems with different mechanisms of action have been identified but in each case the *cas*1 and *cas*2 genes are critical to the initial steps of the process that incorporate the spacer sequences into the CRISPR locus and that later serve as templates for subsequent recognition of matched sequence elements. *Cas6* and *cas9* genes act as nucleases which process CRISPR transcripts in preparation for assembly into ribonucleoprotein particles used for surveillance and ultimately degradation of foreign DNA complementary to a spacer sequence (*Van der Oost et al., 2014*). Variation in the CRISPR-*cas* system complement of *C. fetus* had been reported previously (*Gilbert et al., 2016*; *Van der Graaf-van Bloois et al., 2014*) and this was further examined in this study. Four CFFs had two or more CRISPR loci and a virtually complete complement of *cas* and CRISPR-associated genes representative of a functional CRISPR-*cas* system. Two CFFs, CFF-04-554 and CFF-02A725, as well as all CFV and CFVi isolates lacked *cas1-cas6* genes but retained a modified *cas6, cas10*, a member of the Type III CRISPR-*cas* system, and some other CRISPR-related genes. While these isolates may retain the ability to process spacer transcripts and thereby prevent invasion of elements containing these sequences, the loss of *cas1* and *cas2* would preclude addition of sequences to the CRISPR locus and thereby limit the scope of the system. In those isolates in which the spacer number is significantly reduced the value of this system is clearly highly compromised. Limitations in the functioning of the CRISPR-*cas* system could provide various prophage and transposable elements the opportunity to successfully invade the cell and integrate into the chromosome. While this in itself would not explain the phenotypic changes that have accompanied the emergence of the CFV/CFVi strains, the possibility of random mobile element insertion into the bacterial chromosome and spread within the genome could result in loss of function due to gene deletion or changes in gene expression as well as increased genome instability. Consistent with our observations, it has been reported that the screening of a collection of 102 *C. fetus* isolates for the presence of *cas1* failed to detect this gene in all 62 CFV samples (*Kienesberger et al., 2014*) and it was later suggested that disruption to the CRISPR-*cas* system could be a significant factor contributing to the emergence of CFV/CFVi strains (*Calleros et al., 2017*). We speculate that similar loss of CRISPR-*cas* functionality in CFF isolates, as found here, might initiate a series of genomic modifications to alter their pathogenicity, thus explaining their potential ability to produce infertility in cattle as discussed above. Clearly more work would be needed to substantiate this possibility, but if true, may further blur the distinction between these subspecies in pathogenicity and the role of the two subspecies in BGC.

One notable distinction found between CFF and CFV/CFVi isolates concerns differences with respect to their predicted *dam* gene complement. This gene, first identified and extensively characterised in *Escherichia coli*, encodes DNA adenine methylase responsible for post-replicative adenine methylation at GATC sites along the genome. DNA methylation is an important epigenetic process that can impact aspects of DNA replication, including mismatch repair, as well as gene expression (*Adhikari & Curtis, 2016*; *Casadesús & Low, 2006*). Distinct alleles of a gene predicted to have this activity were found in CFF and CFV/CFVi strains; furthermore, multiple copies of allele 2 of this gene were present in six of the eight CFV/CFVi isolates examined. The structural differences between these *dam* products and potential variation in their expression levels due to variable gene copy number could have significant impact on the extent of GATC methylation of these genomes and, in turn, effect differences in expression of other genes. This aspect of *C. fetus* biology is worthy of further investigation considering that *dam* methylation is believed to impact host-pathogen interactions through modification of virulence gene expression of several bacteria (*Marinus & Casadesus, 2009*).

This report has also explored the plasmid composition of these 14 CF strains as these extrachromosomal elements can contribute genes that impart additional phenotypic features to the organism. Greater numbers of these elements were identified in CFV and particularly CFVi strains, again supporting the concept that these strains are deficient in their ability to limit cellular invasion by foreign DNA. Apart from genes important for replication of these elements, Type IV secretory system genes were the most common. Indeed, a complete T4SS operon was found in CFVi-ADRI1362 P3, thereby complementing the limited gene set present in the chromosome. These genes have undoubtedly been acquired through horizontal transmission from other bacteria (*Kienesberger et al., 2014*) with varying levels of incorporation into the CF chromosome. Moreover, in contrast to prior claims that Type IV secretion system genes were restricted to CFV/CFVi strains, a complete T4SS GI, which corresponds closely to that described previously (*Gorkiewicz et al., 2010*), was found in one of our CFF genomes (CFF-09A980). While these genes are clearly implicated in virulence and pathogenicity their variable complement across these CFs suggests that these functions are likely not restricted to one sub-species.

Critical to this analysis was the comprehensive phenotypic designation of all seven newly analysed isolates described in this study together with their accurate and complete genome characterisation using data derived from multiple short and long read sequencing methods combined with optical mapping for confirmation of genome assembly. The additional seven isolates used for comparison here have also been extensively studied previously, both genetically and phenotypically. These data permitted an evaluation of specific genes identified as either conferring phenotypic differences between the subspecies and/or having utility for subspecies differentiation. The value of the ISfe target for detection of CFV/CFVi strains has already been discussed above. The ability to generate $H_2S$ has been correlated with presence of the L-cys transporter operon comprised of

three genes (*tcyA, tcyB* and *tcyC*) responsible for production of a functional cysteine transporter complex. It was found that all CFV isolates did indeed lack two of these genes while they were retained in all CFVi and CFF isolates. Notably the CFF-00A031 isolate which had been scored as $H_2S$ negative in this study harbored all three genes and would appear to have a functional transporter complex though this has not been verified by expression studies. This observation underscores the challenges in phenotypic evaluation of these highly fastidious organisms and highlights the need for improved genomics-based classification. Indeed it is noteworthy that the CFVi-ADRI1362 isolate has previously been reported as having a CFF phenotype but a genome consistent with its correct designation as described here (*Van der Graaf-van Bloois et al., 2014*). A study exploring the enzymes involved in LPS biosynthesis identified three genes (*glf, mat1* and *wcbK*) which collectively showed good correlation with *C. fetus* subspecies with serotype A CFFs scoring as $glf^+$, $mat1^-$, $wcbK^-$, serotype B CFFs as $glf^-$, $mat1^+$, $wcbK^+$ and all CFVs as $glf^-$, $mat1^+$, $wcbK^-$ (*Kienesberger et al., 2014*). Our analysis of 14 CFs is entirely consistent with this genotype scheme. It has been suggested that the *wcbK* gene product may aid bacterial cell viability in low pH environments and thus facilitate intestinal colonization following ingestion; its absence in all CFVs might in part explain why these strains are most often recovered from the genital tract; however this does not explain how serotype A CFFs navigate the stomach environment. These genetic targets might be further evaluated for their utility in accurate subspecies and biovar identification. As a better understanding of the genetic basis that underpins the distinct pathogenicity of CF isolates is achieved, genetic methods for their discrimination will provide for more accurate and meaningful BGC diagnosis and control.

## CONCLUSION

Our studies overall suggest that *C. fetus* isolates, even those classified as CFF, harbour chromosomes which exhibit some variation in their gene complement and, apart from the retention of a hypervariable region, pathogenicity islands previously described in other studies are not preserved even at the subspecies level. Many intermediate genetic types have evolved as a result of ongoing mobile element acquisition and insertion, genome rearrangement and acquisition or loss of specific genes that collectively may contribute to the variable pathologies caused by these organisms. Variation in the gene complement of extrachromosomal elements further complicates the situation. Deposition of increasing numbers of polished genomes to public databases will help to further clarify the classification of this species and fully understand the genetic basis for its variable pathogenesis.

## ACKNOWLEDGEMENTS

We thank Greg Appleyard for his preliminary studies on the genetic differences between *C. fetus* strains that stimulated this study. We also thank Mary Sheen and Teresa Burke for excellent technical support.

### Funding

This work was supported financially by internal funds from the Canadian Food Inspection Agency and funding from the Genomics Research and Development Initiative of the Government of Canada. The funders had no role in study design, data collection and analysis, decision to publish, or preparation of the manuscript.

### Grant Disclosures

The following grant information was disclosed by the authors:
Canadian Food Inspection Agency.
Genomics Research and Development Initiative of the Government of Canada.

### Competing Interests

The authors declare that they have no competing interests.

### Author Contributions

- Susan A. Nadin-Davis conceived and designed the experiments, analyzed the data, prepared figures and/or tables, authored or reviewed drafts of the paper, and approved the final draft.
- John Chmara performed the experiments, analyzed the data, prepared figures and/or tables, and approved the final draft.
- Catherine D. Carrillo conceived and designed the experiments, authored or reviewed drafts of the paper, and approved the final draft.
- Kingsley Amoako conceived and designed the experiments, analyzed the data, authored or reviewed drafts of the paper, and approved the final draft.
- Noriko Goji performed the experiments, analyzed the data, authored or reviewed drafts of the paper, and approved the final draft.
- Marc-Olivier Duceppe analyzed the data, authored or reviewed drafts of the paper, and approved the final draft.
- John Devenish performed the experiments, analyzed the data, authored or reviewed drafts of the paper, and approved the final draft.

### DNA Deposition

The following information was supplied regarding the deposition of DNA sequences:
Genome data are accessible at GenBank: CP059432 to CP059446 and CP059974.

### Data Availability

Data is available in the Genbank SRA collection and accession numbers are available in the Supplemental Files.

## Supplemental Information

Supplemental information for this article can be found online at http://dx.doi.org/10.7717/peerj.10586#supplemental-information.

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
