# Peer review of "A comparison of fourteen fully characterized mammalian-associated *Campylobacter fetus* isolates suggests that loss of defense mechanisms contribute to high genomic plasticity and subspecies evolution"

_PeerJ, doi:10.7717/peerj.10586_

## Round 0.1 · original submission · Major Revisions

The manuscript will need a lot of work to be accepted, but it is possible.

Reviewer 1 ·

Basic reporting

no comment

Experimental design

no comment

Validity of the findings

no comment

Additional comments

The paper by Nadin-Davis et al describes the complete sequences of three C. fetus subsp fetus (CFF), two C. fetus subsp venerealis (CFV) and two C. fetus subsp venerealis biovar intermedius (CFVi) strains and the comparison between the species including seven other fully sequenced genomes of C. fetus spp. The paper is well written, methods are clear, sequences have been deposited. The availability of more completely sequenced genomes of C. fetus spp. is useful however most of the conclusions have been described before in various papers. I have some minor comments.

Minor comment
1. please add the 7 other C. fetus strains and their genomes to table 1 and table 2. remove table 3. I think their phenotypes are known or at least they can be determined making the comparisons in the text easier to understand while removing a superfluous table.

2. The title suggests that the mechanism for CFV genome plasticity has been found, however a mechanism has not been really described, merely an ineffective CRISPR/CAS system. I would not call that a mechanism. CRISPR/CAS is a mechanism to prevent introduction of foreign DNA, it's absence or ineffectiveness is not a mechanism in CFV to increase plasticity.

3. The start of the discussion mentions that CFF evolved in a human population (Iraola et al 2017). That paper uses a discrete trait analysis using human and animal strains, suggesting that the genomes of ancestral strains of strains infecting humans looked like strains infecting humans now. This is correct. This however does not mean carriage by humans, or "human strains", it means that the strains infecting humans look like other strains having infected humans, but these strains most likely still originated from fecal contaminated bovine or ovine products (e.g. unpasteurised milk). In livestock they did live their life as commensals or potentially even pathogens. There is little to no evidence that these strains evolved in humans as the assumption was flawed to begin with (the flawed assumption being: strains isolated from humans are carried and transmitted by humans to other humans). Clades of CFF are simply extremely undersampled in livestock (carriage) and extremely oversampled in humans (invasive disease). Not surprisingly as CFF is not notifiable in livestock unlike CFV while at the same time CFF in humans is associated with invasive disease, which usually does draw the attention of clinicians and researchers. Perhaps some nuances on the statement "originated in humans" is needed in the start of the discussion as the evidence for that statement is quite thin.

4. Header of table 3 has a typo, but if the strains in table 3 are added to table 1 and 2, this comment can be ignored.

Reviewer 2 ·

Basic reporting

The paper doesn’t emphazise the relevance of subspecies differentiation at the genomic level in the introduction. I think this is very relevant to understand the logics behid the whole experimental design.
L111: inconsistencies in the genetic and phenotypic characteristics of the strains are found. Several papers support it. Some examples are:
Van der Graaf-van Bloois et al. 2014. J Clin Microbiol 52(12): 4183-8; Calleros et al. 2017 J Microbiol Meth 132: 86–94.
This is important to the discussion of the results.
L260: there is an extra parenthesis
L413: please replace “ruminants” for “mammals”
Fig S1, Fig 1 to 4 and TablesS1 and S2: legends for these are missing.
Fig. S1: the quality of the figure is not good. One cannot read the names of the strains

Experimental design

Experimental design is sound and takes into account several aspects of the issue. Technical aspects are well performed. Methods are well described and detailed.

Validity of the findings

Results of the paper are very important and it gives useful information to the understanding of genetic variation and pathogenesis of the species. However, improvement on the discussion is necessary.
The discussion is centered in CRISPR variation, but it fails at citing previous work where some of the hypotheses were already presented (Calleros et al. 2017).
Some of the results are not discussed. Some examples of interesting results that could be discussed:
L224: results from ELISA on serotype are contradictory with previous foundings.
L318-326: the relationship between virulence genes and phenotipic characteristics could be discussed. Some previous work (for example Van Bergen et al. 2005 J Clin Microbiol 43(12): 5888–98; Wang et al. 2013 Vet Microbiol 164(1): 67-76; Kienesbergeret al. 2014 PloS one 9(1): e85491) raises some hypothses about.
Also, subspecies differentiation at the genetic level is a matter of discussion in this species. I think it would be interesting to take this into account.
L393: parA is one of the genes used as a target for diagnostic methods
Several of these results could shed light on the discussion (for example, referred to the genomic evidences of a differentiation between Cff and Cfv) if taken into account.

Additional comments

CGB is a major concern to cattle productive systems. The species C. fetus is increasing its importance as a human pathogen. Accordingly, the analysis of its genetic variation is of paramount significance. Also, it is a fastidious microorganism with very strict growing requirements, so obtaining strains to sequence the genomes with high accuracy and circularized is difficult. This emphasizes the importance of the work presented in the paper.
Results of the paper are very important and it gives useful information to the understanding of genetic variation and pathogenesis of the species. However, improvement on the discussion is necessary.

·

Basic reporting

The manuscript has professional english used throughout.
Lines 102-111 -
-The authors should cite McMillen et al 2006 as the original parA real time assay which was bunked by Spence et al. See - https://jcm.asm.org/content/44/3/938.short. Chaban et al 2012 tried to adapt the parA assay to a SYBR green assay however this target has been shown to not be specific.
The authors should also cite the current OIE - Animal Health standards for CFV diagnostics - this is up to date and summarises the inaccuracies of all currently potential CFV molecular tests. See link https://www.oie.int/fileadmin/Home/eng/Health_standards/tahc/current/chapitre_bovine_genital_campylobacteriosis.pdf
-Also the authors mention biotypes - to this reviewer's knowledge they are 'biovars' - CFV biovar venerealis and CFV biovar intermedius - the authors refer to CFV and CFVi only? please confirm the use of this nomenclature in this manuscript - this reviewer differs.
Line 331 - bound not 'bounded'
Line 387 - within 'most' isolates - quantify exactly rather than generalising
Lines 390-391 - quantify instead of the use of the term 'some'

Experimental design

The experimental design is good -the reporting could do with some additional organisation.
The results section entitled 'genome organisation' needs structure and could include sub-headings. It is otherwise an un-ordered section with many ideas.
The methods are well described.

Validity of the findings

The authors have made several assumptions from their data but lack in interpretation. They have made lists of differences but have not entertained the reason why CFVs are so different to CFFs.
It is not usual to refer to tables and figures in the discussion section.
Table 1 should include the new accession numbers for the Canadian isolates in this study - this is very important and should be included in the manuscript.
The publication cannot be accepted without accession numbers for the new genomes - this is very important. Raw data and annotations need to be reported in the manuscript with accession numbers.

Additional comments

The authors have not really contemplated in the discussion as to why the differences in CFV genomes have evolved? the discussion in the current form is not very intuitive or suggestive.

The publication is worthy of publication - there are several groups who have dominated this field - however the authors have not set themselves apart and the manuscript could be better organised.

Lack of accession numbers for the new genomes is a big issue - this is important for relevance. It is difficult to evaluate without this raw data.

---

## Round 0.2 · Minor Revisions

I consider it has a lot to work on, even though one of the reviewers only asked for minor revisions

Reviewer 2 ·

Basic reporting

Issues have been adressed.

Experimental design

Issues have been adressed.

Validity of the findings

Results of the paper are very important and it gives useful information to the understanding of genetic variation and pathogenesis of the species.

Additional comments

The paper has improved substantially. The discussion section is now more complete and ordered.

However, some major concerns remain:
• In the light of the results, the authors suggest in the conclusion (and I agree with this), that genetic variability of the specie is a continuum and may be subspecies classification is no longer adequate to explain its clinical and ecological behavior. But the problem is this conclusion is not reflect along the discussion, because the treatment of the subspecies separation and the adaptations of the strains to bovines is treated to loosely, confusing some concepts. For example: “While the CFFs contained just one copy of this element, bovine-adapted strains tended to harbour greater copy numbers of these elements and very notably these prophage sequences were often located at the boundaries of sequence rearrangements and inversions supporting the importance of this element type to genome plasticity.” Cff/Cfv classification and adaptations to hosts are not interchangeable concepts. It has been demonstrated in several previous studies. I do recommend to the authors to thoroughly think about this issue and change some statements in the discussion accordingly.
• Some conclusions about bovine adaptation cannot be made because of the absence of comparison with strains isolated from other hosts, so these statements are not supported by the results. There are a human and an ovine strain included in the study, but discussion on this comparison is lacking. Also, all Cfv and Cff strains were collected from the genital tract or aborted fetuses, so it is difficult to extract differences in adaptation from this data.

There are also some minor issues that must be corrected:
• CRISPR-cas system is written in three different ways along the paper, please correct this.
• “Consistent with our observations, screening of a collection of 102 C. fetus isolates for the presence of cas1 failed to detect this gene in all 62 CFV samples (Kienesberger et al. 2014) thus supporting prior suggestions that disruption to the CRISPR-cas system could be a significant factor contributing to the emergence of bovine-adapted strains of C. fetus (Calleros et al. 2017).” Please rewrite to avoid saying that 2017 is prior to 2014.

·

Basic reporting

the article meets the standards

Experimental design

the article meets the standards

Validity of the findings

the article meets the standards

Additional comments

Thank you for undertaking the necessary edtis

---

## Round 0.3 · accepted · Accept

Thank you for improving the manuscript - it is now ready to be published.